# Cross-sectional survey on adult acute kidney injury in Chinese ICU: the study protocol (CARE-AKI)

Zhenyu Yang,[1] Cong Wang,[2] Hongliang Wang,[1] Sicong Wang,[3] Ruijin Liu,[3] Xu Wang,[1] Kaijiang Yu[3]

ZY and CW contributed equally.

## ABSTRACT

**Introduction** Acute kidney injury (AKI) is one of the most serious syndromes in intensive care unit (ICU) patients, and is a mysterious problem in clinical practice worldwide. Due to unknown aetiology and mechanism, awareness of AKI diagnosis and treatment in China varies, resulting in underestimated incidence and poor prognosis. To solve this problem, we design this national survey of AKI in adult ICUs. Various indexes are included and analysed to classify the epidemiology of adult AKI in Chinese ICUs, including AKI aetiology, risk factors, mortality, prognosis, therapeutic strategies and cognition of ICU medical staff.

**Methods** A multicentre, cross-sectional survey, which will involve about 35 hospitals and 6147 patients from 23 provinces, 4 municipalities and 5 autonomous regions, is planned. All patients who meet the inclusion criteria are eligible to apply for enrolment in the study, which cover baseline demographics, clinical performance, and follow-up related to diagnosis and treatment.

**Conclusion** The study is expected to fill the gap between China and developed countries, and to provide a theoretical foundation for developing more scientific and standardised approaches to AKI diagnosis and treatment.

**Ethics and dissemination** Ethical approval was obtained from the ethics committee of Harbin Medical University Cancer Hospital (registration number KY2017-21). The findings of this review will be communicated through peer-reviewed publications and scientific presentations.

**Trial registration number** ChiCTR-EOC-17013133; Pre-results.

## Strengths and limitations of this study

► This is the largest cross-sectional study of acute kidney injury (AKI) to date among adult intensive care unit patients in China.
► This is a comprehensive research programme.
► This is a national study including about 23 provinces, 4 municipalities and 5 autonomous regions.
► The aim of this study is to assess the aetiology, diagnosis, treatment and prognosis of AKI in China.
► This survey is also planned to introduce a medical staff questionnaire which could help to understand knowledge and awareness of AKI among Chinese medical staff.
► Due to the uneven distribution of economic and medical resources in China, our survey might have information biases.

[1]Intensive Care Unit, The Second Affiliated Hospital of Harbin Medical University, Harbin, China
[2]Intensive Care Unit, The First Affiliated Hospital of Heilongjiang, University of Chinese Medicine, Harbin, China
[3]Intensive Care Unit, Cancer Hospital Affiliated to Harbin Medical University, Harbin, China

**Correspondence to**
Dr Kaijiang Yu;
icuyukaijiang@sina.com

## INTRODUCTION

Acute kidney injury (AKI), one of the most common syndromes in critically ill patients, has become a health problem worldwide due to its high mortality and poor prognosis.[1] Due to currently different definitions and unknown mechanisms of acute renal failure, findings of epidemiological investigations are quite different. Previous studies reported an incidence of 5%~20% in critically ill patients.[2] About 13 million of hospitalised patients suffer from AKI, which caused about 1.7 million deaths, and more than 85% of mortality occurred in low-income/middle-income countries. In recent years, the AKI incidence has been increasing in low-income/middle-income countries.[3 4] A national AKI epidemiology study showed the AKI incidence, mortality and misdiagnosis rates were 1%~2%, 12.4% and 74.2% among hospitalised patients in China.[5] In 2012, the Kidney Disease: Improving Global Outcomes (KDIGO) guidelines proposed a new definition and staging criteria for AKI. The new definition of AKI has brought tremendous changes in basic research and conventional clinical practice, thus improving the sensitivity of AKI diagnosis and the ability to more accurately predict the prognosis of AKI. Due to varied medical resource and awareness of AKI, the current situation of AKI diagnosis and treatment remains unclear, especially in Chinese multicentre intensive care units (ICUs). In 2013, the International Society of Nephrology issued a global appeal for the '0 by 25' programme, which is committed to the prevention and treatment of AKI. The goal of the programme is that, by 2025, the preventable mortality of AKI would be reduced to 0.[6] As the largest low-income/middle-income country in the world, China has a large patient population and hospitals have

different medical background. To improve AKI awareness and its prognosis, this study protocol is designed to assess the aetiology, diagnosis, treatment and prognosis of AKI in Chinese ICUs. Therefore, the results are expected to provide important information for future management of Chinese patients with AKI.

## METHODS AND ANALYSIS
### Study design
This study is a national cross-sectional survey involving about 23 provinces, 4 municipalities and 5 autonomous regions. At least one tertiary first-class hospital from each of the provinces is included. Due to the large population and advanced medical resources, two tertiary first-class hospitals are included in Beijing, Shanghai and Guangzhou. As planned, the total number of hospitals will be 35. The total number of patients included in this study is estimated to reach 6147, as calculated using the following formula:

Formula for sample size estimation:

Formula for infinite sample population: $n = \frac{\mu_{\alpha/2}^2 * \pi * (1-\pi)}{\delta^2}$

$\mu_{\alpha/2}^2 = 1.96 \ \pi \ (population\ rate)$   $= 0.02 \ \delta \ (allowable\ error) = 0.01$

$= 1.96^2 * 0.2 * (1 - 0.2) \ / 0.01^2$

$\approx 6147$

### Sampling criteria
#### Inclusion criteria
1. ICU patients receiving consecutive treatments throughout the cross-sectional period.
2. ≥18 years old.

#### Exclusion criteria
1. Patients with chronic kidney disease.
2. Patients who underwent renal transplantation.
3. Patients who underwent routine analysis.

### Date collection
This study is a national, multicentre, prospective, randomised and cross-sectional clinical survey and includes five parts. The standardised forms are created to record related data after patient recruitment (the specific processes are shown in figure 1). Through consensus discussion, the study committee will be elected and will supervise the study procedure. The survey period is intended to be from 08:00 on the first day to 08:00 on the 31st day of the study.

### Patient demographics and clinical performance
1. Basic information.
2. Main diagnosis in the ICU and basic creatinine values, and others.

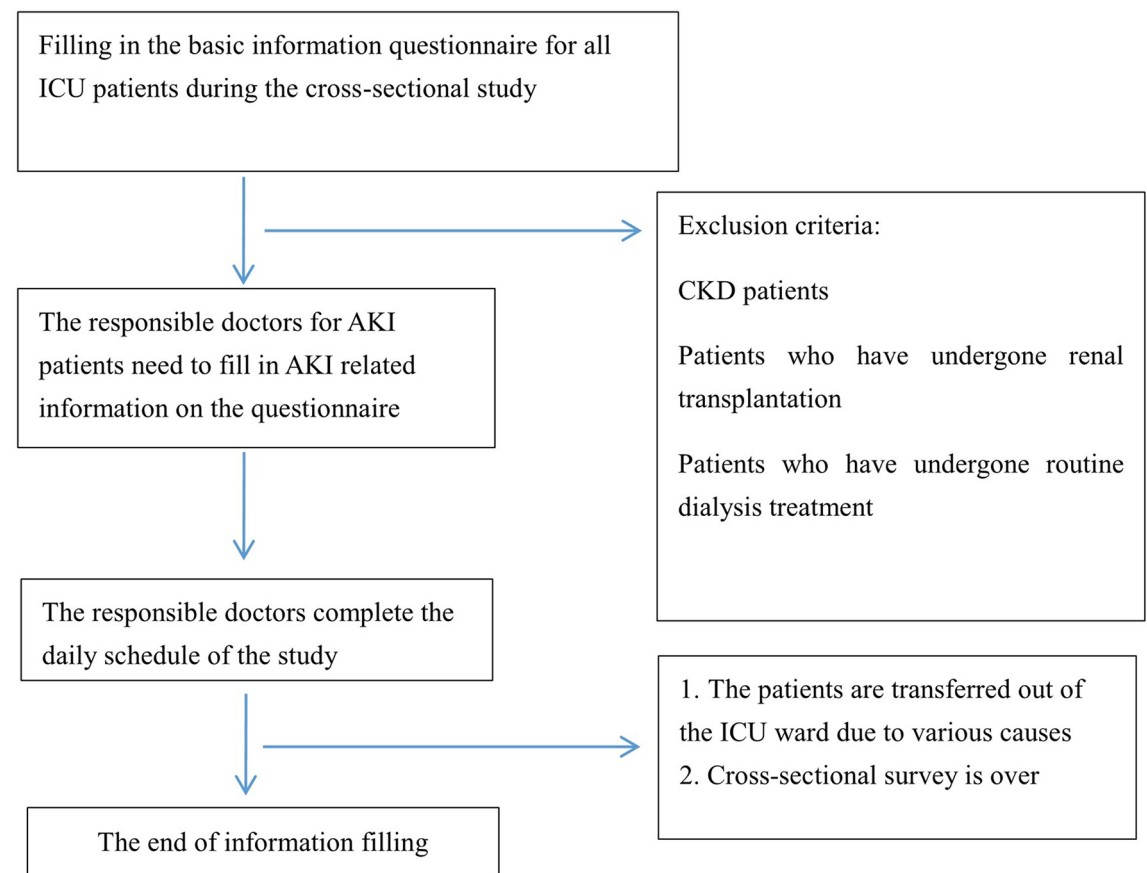

**Figure 1** This study is a national, multicentre, prospective, randomised and cross-sectional clinical survey in China. Thus, a high-quality, cross-sectional survey is needed, and the process of performing this study can be found in the figure, which will be separated into four parts. AKI, acute kidney injury; CKD, chronic kidney disease; ICU, intensive care unit.

3. Acute Physiology And Chronic Health Evaluation (APACHE II score), Glasgow Coma Scale (GCS score) and Sequential Organ Failure Assessment (SOFA score) (the highest score within a 24-hour period).

### Investigation of patients with AKI

1. Time of AKI diagnosis, diagnosis criteria and staging.
2. Nephrotoxin drugs used prior to the diagnosis of AKI.
3. Aetiology of AKI (prerenal, renal, postrenal).
4. Life support and its modes.
5. Application of vasoactive drugs and the prescription of renal replacement therapy (RRT) treatment.
6. Severity and prognosis of AKI.

### ICU facility and infrastructure

The general condition of participating ICUs is recorded, including ICU type, number of beds, patient population, blood purification machines, RRT strategy and the annual cases in the ICU.

### Survey for ICU doctors

The participating doctors are enquired for their personal information, clinical experience, and especially their knowledge/awareness about AKI, including diagnosis, treatment and kidney replacement treatment.

### Survey for ICU nurses

The participating nursing staff are enquired for their personal information, clinical experience, and especially their knowledge/awareness about AKI and blood purification machines.

### Measurements and statistical analysis

1. The results are mainly compared for these indexes: AKI aetiology, incidence, mortality, ICU cost, total hospitalisation and prognosis.
2. Risk factors are screened to evaluate AKI prognosis, including diuretics, blood infusion, infusion of crystalloid or colloid, intra-abdominal pressure and others.
3. Among patients with AKI, patients with sepsis are filtered to apply the subgroup analysis, in order to classify the relationship between severity and prognosis of AKI.
4. ICU facility, team composition, medical awareness of staff and AKI guideline implementations are surveyed to illuminate the district variance due to different medical resources.
5. Current application and prescription of RRT treatments are counted among patients with AKI and evaluated for their efficiency in China, in order to foster standardised clinical protocol. Further research on continuous blood purification will be carried out on this basis.

### Protection of patient rights

All participating patients are informed and sign a written consent. All patient information is kept confidential.

## DISCUSSION

Kidney is an important excretory and endocrine organ in the body and is critical to maintaining body homeostasis and normal metabolism. In ICU, underlying diseases such as severe infection, multiple trauma and a variety of shock symptoms are highly likely to induce AKI or exacerbate the pre-existing condition of AKI. Kidney dysfunctions can further aggravate patients' conditions and affect their prognosis. Prevention, diagnosis and treatment of AKI are often critical for public health worldwide. The AKI aetiology and mechanisms are still unknown. There is also no consensus standards in terms of AKI early diagnosis, treatment strategy and RRT. Due to the different definitions among countries, AKI incidence was reported to be significantly varied.[7] Most recent studies were from developed countries, whereas reports are scarce in low-income/middle-income countries. But the AKI incidence is increasing and underestimated in low-income/middle-income countries.[7–9] Many researches pointed out the epidemiological data were also not reliable for those single-centre studies in Africa, Asia and Latin America.[10–12] For example, a study in Saudi Arabia in 2003 showed that the AKI incidence was 2.3/1000 of hospitalised patients,[13 14] while a single centre in India reported an AKI incidence of 6.6/1000 among hospitalised patients in 2007.[15] In South-East Asia, due to the monsoon climate, malaria, leptospirosis and gastrointestinal tract infection, the AKI incidence was up to 18%~24%.[16] A national and multi-centre AKI epidemiology study was conducted in China in 2015, which indicated the AKI incidence, mortality and misdiagnosis rates were 1%~2%, 12.4% and 74.2% among hospitalised patients.[5] But the current situation of AKI diagnosis and treatment in Chinese ICUs remains unclear. This survey study mainly covers several critical aspects for managing AKI: diagnosis, aetiology, life support and monitoring, prognostic indicators, and medical staff's awareness. Through our findings, we might answer these questions and we expect to provide effective and standardised approaches for future AKI management in Chinese ICUs.

### Diagnosis criteria and the aetiology of AKI

The distinct criteria for early AKI result in different times of accurate diagnosis and greatly affect subsequent clinical intervention. The time and criteria of AKI diagnosis were investigated among multicentre ICUs, including Risk/Injury/Failure/Loss/End-stage (RIFLE) in 2005, the Acute Kidney Injury Network (AKIN) in 2007 and the KDIGO in 2012. Their sensitivity, specificity and clinical effects are compared to optimise clinical practice in AKI diagnosis. Referring to the KDIGO guidelines, the aetiology of AKI is recorded and compared as required, such as history of infection, severe disease, shock, burn, trauma, cardiac surgery (especially surgeries involving extracorporeal circulation) and large non-cardiac surgery, nephrotoxic drugs, contrast agents, and exposure to poisonous plants or animals.[17] These will illuminate general AKI

aetiology and regional differences, which are important for medical awareness and early prevention of AKI.

## ICU life support concerned with AKI

There are a lot of controversial findings about life support approaches to induce or aggravate AKI in ICUs. For this cross-sectional study, we intend to collect larger data to figure out their effects on AKI, especially for mechanical ventilation, diuretics, vasoactive drugs, renal perfusion and RRT. To assess the effects of abdominal pressure on AKI, ICU patients with mechanical ventilation are enrolled. The mode of mechanical ventilation is recorded. The abdominal pressure in patients with AKI should be monitored during the first 24 hours after admission. In 2012, the KDIGO guidelines no longer recommended routine use of diuretics. Loop diuretics were previously reported to improve renal function, removing the block particles and increasing renal blood flow.[18] [19] Further studies failed to verify the beneficial effects, but showed that such drugs increased mortality.[20] In our survey, the diuretics prescription is recorded in detail to further clear the confusion and correct medicine knowledge, such as diuretic types, dosage, drug delivery and haemodynamic indexes. Some investigations are done to study the inconsistent effects of vasoactive drugs on renal function.[21] [22] Different drugs are compared in AKI cases, including dopamine, norepinephrine, epinephrine and dobutamine.

Renal perfusion and resuscitation resulted in different outcomes by varied fluid type.[23] [24] Hydroxyethyl starch might aggravate the incidence and severity of AKI, and even increase mortality in severe AKI cases. In critically ill patients, hypertonic chloride liquid might induce or aggravate hyperchloraemia and metabolic acidosis, and eventually reduce glomerular filtration rate by renal vasoconstriction. The fluid type of resuscitation is required to collect, including blood products ,crystalloid or colloid. Also the volume of intake/output fluid in 24 hours is required. RRT treatment is critical to treating patients with AKI, but its prescription and application guidelines are with no consensus.[25] Through our large-sample multicentre study, we record the different strategies and outcomes of RRT treatment to foster standardised clinical approaches.

## Prognostic indicators

The AKI mortality from different countries is affected by medical status, economic level and geographical location.[26] If the different social and economic factors of these countries were considered in the statistical analysis, mortality in different groups was similar.[27] In this study, we focus on the prognosis, staging (KDIGO) and clinical outcomes of patients with AKI, such as in-hospital mortality, 28-day mortality and long-term RRT. The ICU facility, local economy and personal characteristics are also introduced to adjust our statistical calculation. More importantly, the difference in AKI prognosis will foster individual health policy and treatment strategy in China.

## Medical staff awareness

AKI knowledge and awareness are critical to clinical practice. In this cross-sectional survey, the questionnaires are introduced to objectively evaluate medical acquisition of medical staff. The team composition of doctors and the nurse ratio are also recorded. Through data mining, the results can inspire further education and team optimisation of ICU staff. The varied finding on doctors' knowledge will be calculated to solve analysis bias and evaluate accurate AKI epidemiology in China.

This is the first national survey on the epidemiology of AKI among ICUs in China. As planned, one representative hospital is recruited from each province/municipality or autonomous region. The findings would reflect the real status of AKI throughout China, help to establish standardised guidelines and promote individual optimisation of ICU AKI treatment. This survey also introduces a medical questionnaire for large staff population, which can be used to understand knowledge and awareness of AKI. Through this survey, we expect to improve our clinical practice in the management of AKI in China. These data would also inspire further studies in other low-income/middle-income countries.

## Study status

All preliminary work has finished. The data collection of this cross-sectional survey will be carried out in 2018.

**Contributors** No person other than the authors listed here has contributed significantly to the manuscript preparation. All persons listed here as authors contributed to the preparation of the manuscript, and the International Committee of Medical Journal Editors' (ICMJE) criteria for authorship were met. Specifically, the following contributions were made by the respective authors: KY: acts as the project's principal investigator, and contributed to study conception and design; agree to be responsible for all aspects of the study; approved the final draft before its publication. ZY, CW: collected and studied a large number of relevant references; drafted the study report and revised it; approved the final draft before its publication. HW, SW, RL, XW: collected important background information related to the study; contributed substantially to the acquisition, analysis and interpretation of data; drafted the study report and revised it; approved the final draft before its publication.

**Funding** This project is funded by the National Natural Science Foundation of China (no 81571871 and 81770276).

**Competing interests** None declared.

**Patient consent** Guardian consent obtained.

**Ethics approval** The study was approved by the ethics committee of the participating hospital (Harbin Medical University Cancer Hospital) in China.

**Provenance and peer review** Not commissioned; externally peer reviewed.

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
