## [Reviewer comments · BMJ Open]

ARTICLE DETAILS

TITLE (PROVISIONAL)	A cross-sectional survey on adult acute kidney injury in Chinese ICU : the study protocol (CARE-AKI)
AUTHORS	Yang, Zhenyu; Wang, Cong; Wang, Hongliang; Wang, Sicong; Liu, Ruijin; Wang, Xu; Yu, Kaijiang

VERSION 1 – REVIEW

REVIEWER	Zhi-Yong Peng University of Pittsburgh Medical Center USA
REVIEW RETURNED	17-Dec-2017

GENERAL COMMENTS	This is the largest survey in Chinese ICU, which will provide the data about the epidemiology of AKI and current situation of practice. This study is interesting , clinical relevant and has important implications. However, the authors need to sort out the following concerns before this manuscript is accepted: 1.Introduction: need to add sentences on the current practice in China and the gap between China and the developed countries;2. Methods: need to be re-organized based on the study population,sampling criterion,procedures of data collection, measurements and statistical analysis;3. Discussion: need to shorten, and focus on the strength/limitation and future implication of the project;4.Some English grammar errors/ typos need to be revised
--

REVIEWER	Pr Thomas Rimmelé Anesthesiology and Intensive care Medicine Edouard Herriot Hospital Lyon France
REVIEW RETURNED	18-Dec-2017

GENERAL COMMENTS	The manuscript entitled "A cross-section survey on adult acute kidney injury in Chinese ICU: the study protocol" by Zhenyu Yang et al. is the protocol description of a large multicenter observational epidemiologic study of AKI in China. This study will be launched in January 2018. In my opinion, the manuscript is overall well-written and the study design is obviously interesting since this work will become the largest cross-sectional AKI study in China so far. I have several comments that should be taken into consideration before potential publication. Major comments - For a cross-sectional study, timing is very important. The timing
---

	information, I mean by timing the precise period when the study will be conducted, should therefore be given in the protocol, in the abstract and in the body text (material and methods section). It is currently written that the study will last one month, from 8 AM on the first day to 8 AM on the 31 st day but could the authors provide the exact period of time? In the same line, will all centers be entering the data during the same period of time?  - This is a large study involving 35 hospitals. Information regarding funding should be available. It is currently said no funding statement. It is surprising since it is difficult to imagine that there is no budget dedicated for this work. - Abstract: Description of the study design should be more developed. For example, it is not said in the abstract that the knowledge of AKI among Chinese doctors will be assessed. - It is my understanding that some information regarding RRT such as the reason why RRT is initiated will be recorded. Do you plan to also record additional parameters from the RRT prescription (flow rates, anticoagulation, RRT dose, catheters information...). If yes, the authors should highlight this in the manuscript. - Information that will be recorded could be given in a more structured way. The authors should precisely list all information the scientific community will obtain thanks to this study. - There are some grammar and syntax errors throughout the manuscript; I suggest that a colleague for whom English is a first language revises the manuscript Minor comments  - Title: cross-sectional instead of cross-section - Does this study have a name? If not, it could be useful to give it one - Introduction, first sentence: AKI is not a “disease”, it is a “syndrome”. Please correct. - Methods and analysis: The authors should be very clear on the fact that the total number of patients enrolled in this study is, at this stage, just an estimation. I would therefore go for: “The total number of patients included in this study is estimated to reach 6147, as calculated from the following formula. - Why do you use the past tense for this manuscript? The study has not been performed yet! I would recommend to use the present tense. This study is a national..... The total number of hospitals is 35..... - References should be all formatted in the same way. I wish the investigators all the best for this work.
--	---

VERSION 1 – AUTHOR RESPONSE

Dear editor and reviewers:

Thanks for the great review about Manuscript ID bmjopen-2017-020766 entitled "A cross-section survey on adult acute kidney injury in Chinese ICU : the study protocol" . I have revised the manuscript as required.

Reviewer: 1

Reviewer Name
Zhi-Yong Peng

1. Introduction: need to add sentences on the current practice in China and the gap between China and the developed countries;

Answers : Thanks for Prof. Peng's comments, we have revised our manuscript and add more description about current practice in China and its gap.

2. Methods: need to be re-organized based on the study population, sampling criterion, procedures of data collection, measurements and statistical analysis;

Answers : The design of manuscript structure has been modified as required.

3. Discussion: need to shorten, and focus on the strength/limitation and future implication of the project;

Answers: We have rewritten our draft and provide more focus discussion about our study aim and its limitation.

4. Some English grammar errors/ typos need to be revised

Answers: We are sorry for the poor English. The expression has been revised by some experts.

Reviewer: 2

Reviewer Name

Pr Thomas Rimmelé

Major comments

- For a cross-sectional study, timing is very important. The timing information, I mean by timing the precise period when the study will be conducted, should therefore be given in the protocol, in the abstract and in the body text (material and methods section). It is currently written that the study will last one month, from 8 AM on the first day to 8 AM on the 31 st day but could the authors provide the exact period of time? In the same line, will all centers be entering the data during the same period of time?

Answers: Thanks for the suggestion. The period for real study will be discussed among different centers. With the consensus agreement, we will start and collect the data strictly according to the protocol.

- This is a large study involving 35 hospitals. Information regarding funding should be available. It is currently said no funding statement. It is surprising since it is difficult to imagine that there is no budget dedicated for this work.

Answers: The funding information has been added.

- Abstract: Description of the study design should be more developed. For example, it is not said in the abstract that the knowledge of AKI among Chinese doctors will be assessed.

Answers: The abstract has been revised as required. More study design and AKI status in China has been described.

- It is my understanding that some information regarding RRT such as the reason why RRT is initiated will be recorded. Do you plan to also record additional parameters from the RRT prescription (flow rates, anticoagulation, RRT dose, catheters information...). If yes, the authors should highlight this in the manuscript.

Answers: Thanks for the suggestion. We have considered to record the RRT prescription in our protocol to classify the treatment difference.

Also we have another cross-sectional survey that collects information about RRT in detail. The name of study is “A multi-center cross-sectional study on Blood Purification of ICU adult patients in China”.

- Information that will be recorded could be given in a more structured way. The authors should precisely list all information the scientific community will obtain thanks to this study.

Answers: The study design and record protocol has been revised, and we hope it will help more international colleagues to promote their clinical researches.

- There are some grammar and syntax errors throughout the manuscript; I suggest that a colleague for whom English is a first language revises the manuscript

Answers: We are sorry for the poor English. The expression has been revised by some native experts.

Minor comments

- Title: cross-sectional instead of cross-section

- Does this study have a name? If not, it could be useful to give it one

- Introduction, first sentence: AKI is not a “disease”, it is a “syndrome”. Please correct.

- Methods and analysis: The authors should be very clear on the fact that the total number of patients enrolled in this study is, at this stage, just an estimation. I would therefore go for: “The total number of patients included in this study is estimated to reach 6147, as calculated from the following formula.

- Why do you use the past tense for this manuscript? The study has not been performed yet! I would recommend to use the present tense. This study is a national..... The total number of hospitals is 35.....

- References should be all formatted in the same way.

Answers: We are sorry for the wrong expression. We have revised the manuscript as required.

And we have given a name for our study :CARE-AKI .

VERSION 2 – REVIEW

REVIEWER	Zhiyong Peng University of Pittsburgh Medical Center, Pittsburgh, PA 15213, USA
REVIEW RETURNED	08-Mar-2018

GENERAL COMMENTS	I suggest to add the strength of the study and limitation of the study in the section of discusiion. I agree other revision.
--

REVIEWER	Pr Thomas Rimmelé Anesthesiology ash Critical Care Medicine Edouard Herriot Hospital Hospices Civils de Lyon France
REVIEW RETURNED	23-Mar-2018

GENERAL COMMENTS	The authors have addressed correctly my concerns
--